# Co-existing feedback loops generate tissue-specific circadian rhythms

J Patrick Pett[1], Matthew Kondoff[3], Grigory Bordyugov[3], Achim Kramer[2], Hanspeter Herzel[3]

Gene regulatory feedback loops generate autonomous circadian rhythms in mammalian tissues. The well-studied core clock network contains many negative and positive regulations. Multiple feedback loops have been discussed as primary rhythm generators but the design principles of the core clock and differences between tissues are still under debate. Here we use global optimization techniques to fit mathematical models to circadian gene expression profiles for different mammalian tissues. It turns out that for every investigated tissue multiple model parameter sets reproduce the experimental data. We extract for all model versions the most essential feedback loops and find auto-inhibitions of period and cryptochrome genes, *Bmal1–Rev-erb-α* loops, and repressilator motifs as possible rhythm generators. Interestingly, the essential feedback loops differ between tissues, pointing to specific design principles within the hierarchy of mammalian tissue clocks. Self-inhibitions of *Per* and *Cry* genes are characteristic for models of suprachiasmatic nucleus clocks, whereas in liver models many loops act in synergy and are connected by a repressilator motif. Tissue-specific use of a network of co-existing synergistic feedback loops could account for functional differences between organs.

## Introduction

Many organisms have evolved a circadian (~24 h) clock to adapt to the 24-h period of the day/night cycle (1). In mammals, physiological and behavioral processes show circadian regulation including sleep–wake cycles, cardiac function, renal function, digestion, and detoxification (2). In most tissues, about 10% of genes have circadian patterns of expression (3, 4). Surprisingly, the rhythmicity of clock-controlled genes is highly tissue specific (4, 5, 6).

Circadian rhythms are generated in a cell-autonomous manner by transcriptional/translational feedback loops (7) and can be monitored even in individual neurons (8) or fibroblasts (9).

Ukai and Ueda (10) depict the mammalian core clock as a network of 20 transcriptional regulators (10 activators and 10 inhibitors) acting via enhancer elements in their promoters such as E-boxes, D-boxes, and retinoic acid receptor-related orphan receptor elements (RREs). Because many of these regulators have similar phases of expression and DNA binding (11, 12), the complex gene regulatory network has been reduced by Korenčič et al (6) to just five regulators representing groups of genes: the activators *Bmal1* and *Dbp* and the inhibitors *Per2*, *Cry1*, and *Rev-Erba* (Fig 1A and B).

Even this condensed network contains 17 regulations constituting multiple negative and positive feedback loops (13). To generate self-sustained oscillations, negative feedback loops are essential (14, 15). Originally, the self-inhibitions of the period and cryptochrome genes have been considered as the primary negative feedback loops (16). Later, computational modeling (17) and double-knockout experiments suggested that the *Rev-Erb* genes also play a dominant role in rhythm generation (18). Recently, it has also been shown that a combination of three inhibitors forming a repressilator (19) can reproduce expression patterns in the liver, adrenal gland, and kidney (13).

Despite many experimental and theoretical studies, major questions remain open: What are the most essential feedback loops in the core clock network? Do dominant loop structures vary across tissues?

Here, we use global optimization techniques to fit our five-gene model to expression profiles in different mammalian tissues (adrenal gland, kidney, liver, heart, skeletal muscle, lung, brown adipose, white adipose, suprachiasmatic nucleus (SCN), and cerebellum) (3). We find that for any given tissue, multiple parameter sets reproduce the data within the experimental uncertainties. By clamping genes and regulations at non-oscillatory levels (13), we unravel the underlying essential feedback loops in all these models. We find auto-inhibitions of the period and cryptochrome genes, *Bmal1–Rev-erb-α* loops, and repressor motifs as rhythm generators. The role of these loops varies between organs. For example, in the liver, repressilators dominate, whereas *Bmal1–Rev-erb-α* loops are found in the heart. Clustering of the model parameter sets reveals tissue-specific loop structures. For example, we rarely find the repressilator motif in the brain, heart, and muscle tissues because of the earlier phases and small amplitudes of *Cry1*. We discuss that the co-existence of functional feedback loops increases robustness and flexibility of the circadian core clock.

[1]Institute for Theoretical Biology, Humboldt-Universität zu Berlin, Berlin, Germany   [2]Laboratory of Chronobiology, Charité-Universitätsmedizin Berlin, Berlin, Germany   [3]Institute for Theoretical Biology, Charité-Universitätsmedizin Berlin, Berlin, Germany

Correspondence: h.herzel@biologie.hu-berlin.de

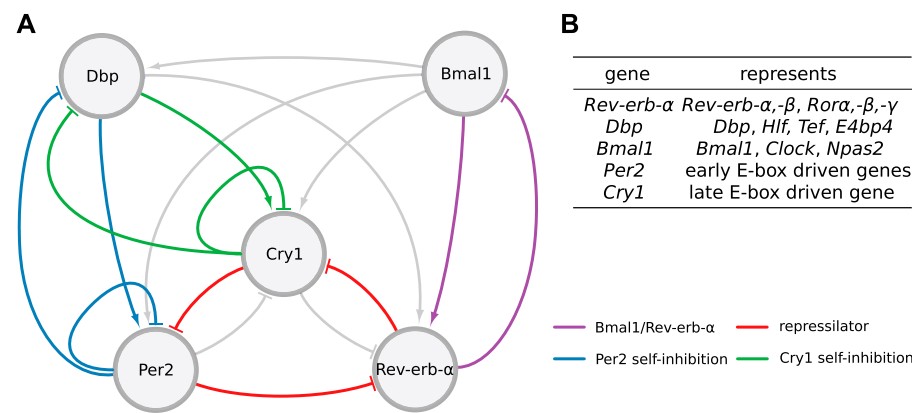

**Figure 1. Network of the core clock model.**
**(A)** The graph comprises 7 activations and 10 inhibitions forming several negative feedback loops. Four loops that are mainly discussed in the literature and were most often found by our analysis are marked in different colors. Note that for Per2 and Cry1 auto-inhibitions also, extensions via the gene Dbp are counted. **(B)** Table of genes represented by each variable of the model.

| gene | represents |
|---|---|
| *Rev-erb-α* | *Rev-erb-α,-β, Rorα,-β,-γ* |
| *Dbp* | *Dbp, Hlf, Tef, E4bp4* |
| *Bmal1* | *Bmal1, Clock, Npas2* |
| *Per2* | early E-box driven genes |
| *Cry1* | late E-box driven gene |

— Bmal1/Rev-erb-α   — repressilator
— Per2 self-inhibition   — Cry1 self-inhibition

# Results

### A five-gene regulatory network represents most essential loops

Here, we derive a gene regulatory network that can be fitted successfully to available transcriptome, proteome, and ChIP-seq data. We will use the model to explore tissue-specific regulations.

Many circadian gene expression profiles for mouse tissues are available (5, 20, 21). Here, we focus on data sets from tissues spanning 48 h with a 2-h sampling (3). These comprehensive expression profiles are particularly well suited to study tissue differences. For mouse liver also, proteome (22) and ChIP-seq data (11, 23, 24) are available with lower resolution.

Using global parameter optimization, we fit tissue-specific model parameters directly to the gene expression profiles of Zhang et al (3). Proteome and ChIP-seq data are primarily used to specify reasonable ranges of the delays between transcription and the action of activators and repressors. The ranges of degradation rates have been adapted to large-scale studies measuring half-lifes of mRNAs (25, 26) and proteins (27).

Quantitative details of activation and inhibition kinetics are not known because of the high complexity of transcriptional regulation. The transcriptional regulators are parts of MDa complexes (28) including histone acetyltransferases and histone deacetylases (29). Details of DNA binding, recruitment of co-regulators, and histone modifications are not available (30). Thus, we use heuristic expressions from biophysics (31) to model activation and inhibition kinetics. Exponents represent the number of experimentally verified binding sites (32) (Supplementary Information 1), and the parameters were assumed to be in the range of the working points of regulation.

To justify the topology of our reduced gene regulatory network, we analyze the amplitudes and activation phases of all the 20 regulators described in Ukai and Ueda (10) (Fig 2). Repressor phases were inverted by 12 h to reflect the maximal activity and allowing direct comparison with activators.

Fig 2 shows that the five genes binding to RREs and the four genes binding to D-boxes cluster at specific phases. Consequently, we represent these regulators by selected genes with large amplitudes: *Rev-erb-α* and *Dbp*. Because the other RRE and D-box regulators peak at similar or directly opposed phases, their additional regulation can be taken into account by the fitting of activation and inhibition parameters.

The regulation via E-boxes is quite complex (11, 30, 33, 34). In addition to the activators *Bmal1* and *Bmal2*, we have their dimerization partners *Clock* and *Npas2* and their competitors *Dec1* and *Dec2*. Furthermore, there are the early E-box targets *Per1*, *Per2*, *Per3*, and *Cry2* and the late gene *Cry1*. We model this complicated modulation by three representative genes: *Bmal1* as the main activator and *Per2* and *Cry1* as the early and late E-box target, respectively.

In summary, the reduced gene regulatory network consists of five genes and 17 regulations (Fig 1). All regulations and the number of binding sites have been confirmed by several experimental studies discussed in detail in (32). Interestingly, liver proteomics (22) and ChIP-seq data are consistent with morning activation via *Bmal1*, evening activation by *Dbp*, and sequential inhibition by *Rev-erb-α*, *Per2*, and *Cry1*. Recent detailed biochemical experiments support the notion that there are distinct inhibition mechanisms associated with *Per2* and *Cry1* (30). The essential role of the late *Cry1* inhibition has been stressed also by Ukai-Tadenuma et al (35) and Edwards et al (36).

As discussed above, our model is fitted directly to mRNA time series collected for different tissues at 2-h intervals for a total duration of 2 d. The transcriptional/translational loops are closed by delayed activation or repression realized by the corresponding proteins. Because most quantitative details of posttranscriptional modifications, complex formations, nuclear import, and epigenetic regulations are not known, we simplify all these intermediate processes by using explicit delays. Thus, we describe the core clock network by five delay-differential equations with 34 kinetic parameters (see Supplementary Information 1 for the complete set of equations).

The model constitutes a strongly reduced network that approximates the highly complex protein dynamics by delays. Inhibition strengths are represented by a single parameter, whereas modeling of activation requires two parameters: maximum activation and threshold levels. While keeping in mind the proposed simplifications, the resulting tissue-specific models can still be regarded as a biologically plausible regression of the underlying biological dynamics.

In Fig 3A, we show examples of simulations fitted to the corresponding gene expression patterns. After successful parameter optimization as described below, the differences between data and models are comparable with experimental uncertainties quantified by comparing different studies (3, 21, 37) and by studying the

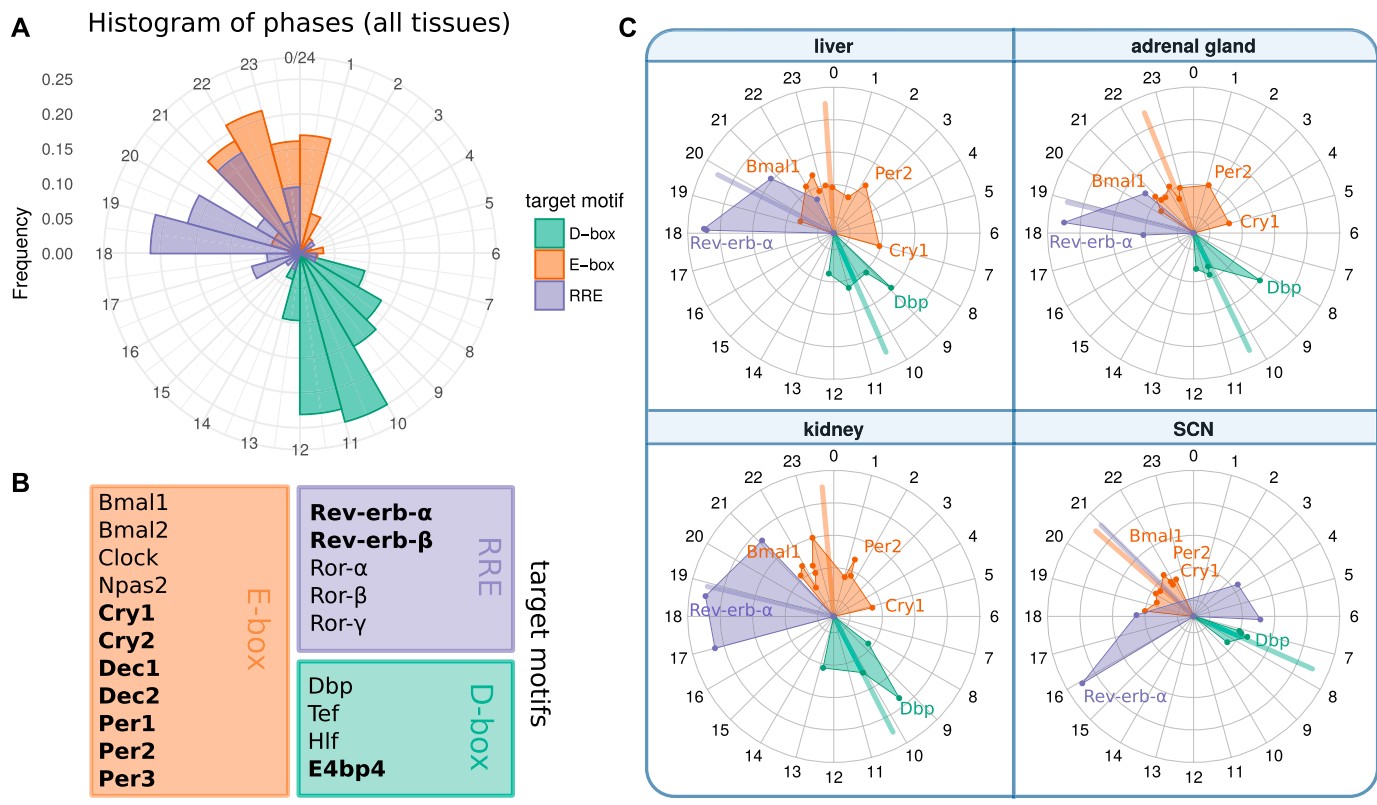

**Figure 2. Circular plots of 20 regulators reveal redundancies and serial inhibition (10, 12).**
They represent peak phases of mRNA expression in multiple tissues (3). Note that repressor phases were inverted by 12 h to allow direct comparison with activators. **(A)** Histogram of the phase distribution over all tissues. **(B)** List of genes represented in circular plots and their corresponding target motifs. Repressors are marked in bold. **(C)** Phases of core clock genes in selected tissues. Colored lines correspond to the circular mean of the respective groups. Amplitudes are linearly scaled. The differences between SCN and other tissues are particularly notable (e.g., the earlier Cry1 peak). Antagonistic regulations of Rev-erb and Ror in the SCN can be modeled by reduced inhibition strength of Rev-erb-α.

differences between the first and second day of the expression profiles by Zhang et al (3) (Supplementary Information 3).

### Vector field optimization (VFO) improves model fitting

To investigate whether our reduced gene regulatory network can reproduce tissue-specific data, (3) we developed a pipeline for global parameter optimization and analysis (scheme in Fig 3B). We applied the pipeline multiple times to each tissue-specific expression profile, allowing us to compare optimized model parameters between tissues. Table 1 lists the number of optimization runs for 10 analyzed tissues. Four tissues (liver, SCN, adrenal gland, and kidney) are discussed in more detail in the following sections, whereas results for others can be found in Supplementary Information 2 and 5.

The agreement of model simulations and experimental mRNA time courses (3) were measured by a scoring function. In this

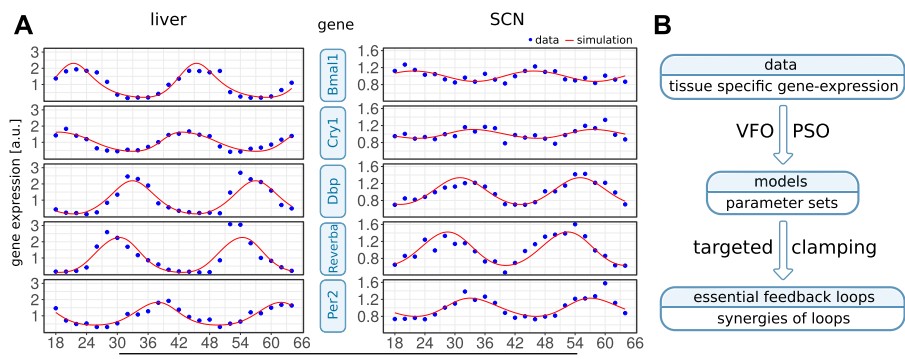

**Figure 3. (A)** Example time series for data and simulation. One fit in liver (left) and one fit in the SCN (right) are shown. Expression levels are normalized to the mean values. The liver fit has a score of 0.01 and involves the *Bmal1–Rev-erb-α*, repressilator, and *Cry1* loops, whereas the SCN fit scores 3.36 and involves the *Bmal1–Rev-erb-α*, *Per2*, and *Cry1* loops. Note the smaller amplitudes and the early *Cry1* phase in the SCN. **(B)** Workflow of the analysis. Multiple optimized parameter sets are obtained from each tissue-specific data set. Then essential loops are identified in the respective models.

**Table 1. Number of optimization runs per tissue and average score.**

|  | Adrenal gland | Kidney | Liver | Heart | Skeletal muscle | Lung | Brown adipose | White adipose | SCN | Cerebellum |
|---|---|---|---|---|---|---|---|---|---|---|
| Number of runs | 100 | 93 | 57 | 57 | 58 | 31 | 62 | 45 | 153 | 58 |
| Runs with score < 10 | 66 | 59 | 52 | 44 | 35 | 21 | 36 | 22 | 46 | 39 |
| Mean score | 3.84 | 4.48 | 1.58 | 3.74 | 5.17 | 3.04 | 4.00 | 3.24 | 7.21 | 3.99 |
| Figure | 2, 5, 6, 7, 8 | 2, 5, 6, 7, 8 | 2–8 | 8, S2, S5 | 8, S2, S5 | 8, S2, S5 | 8, S2, S5 | 8, S2, S5 | 2–8 | S2, S5 |

Four tissues (adrenal gland, kidney, liver, and SCN) are mainly discussed in the main text and others are described in supplements as indicated in the last row.

function period, relative phases and fold changes of measured gene transcripts are taken into account. The complete scoring function is given in Supplementary Information 3.

Model parameters are chosen by global optimization, such that the score obtained by our scoring function is minimal. The optimization method approaches a local minimum in a high dimensional parameter space, and thus, final scores of each run depend on the starting conditions. We only used model fits with scores lower than a chosen threshold of 10 for further analyses. A cutoff of 10 reflects deviations that are within the experimental uncertainties according to our tolerance values (Supplementary Information 3). Interestingly, the fractions of optimization runs with scores lower than 10 vary across tissues. Whereas the largest number of successful runs is found for liver data (about 90%), for the kidney and adrenal gland about 2/3 and for SCN only 1/3 of the runs yield good scores below the chosen threshold.

Allowed ranges for parameters were defined to restrict the search space. Although delays and degradation rates are optimized within biologically plausible ranges around experimentally measured values, for activation and inhibition strengths no such measurements are available. Therefore, we define ranges based on oscillation mean levels and corresponding to the working points of regulations, that is, ranges in parameter space in which regulation strengths vary most.

Global optimization is performed with particle swarm optimization (PSO) (38). A number of particles—each representing one parameter combination—are initialized randomly using Latin hypercube sampling (39) and moved around in the parameter space with velocities changing according to both their individual and their neighbor's known best location. The movements are conducted for a number of iterations while velocities decrease and particles converge to an optimum.

We improve global optimization by identifying good starting conditions. To this end, we devise a strategy which we here call "vector field optimization." Our algorithm makes use of experimentally-derived time courses for model variables and their mathematical description in terms of differential equations. From the data, we can approximate the time derivatives together with the right-hand sides of our model equations (Fig 4A and Supplementary Information 4). By minimizing the differences, we obtain initial values of model parameters. This step does not require simulation, but already yields parameter combinations that account for much of the differences between time courses. For example, the known antiphase oscillations of *Bmal1* and *Rev-erb-α* can be generated with a *Bmal1* delay of about 6 h. Even though the overall search space of this parameter is the interval from 0 to 6 h,

VFO leads to initial delay values close to 6 h (see Supplementary Information 4 for details).

VFO is performed using a bounded gradient method to ensure that solutions lie within the parameter limits. Starting points for the gradient method are chosen randomly. We tested whether VFO improves the scores of model fits. Indeed, we are able to find significantly more good fits for the liver and SCN than with PSO alone (Fig 4B). Notably, in the SCN, it was difficult to reach scores lower than 10 without previous application of VFO.

### Clamping reveals essential loops

Using global optimization, we found for all 10 tissue-specific expression profile (3) parameter sets that reproduce the data within experimental uncertainties (Supplementary Information 2 and 3).

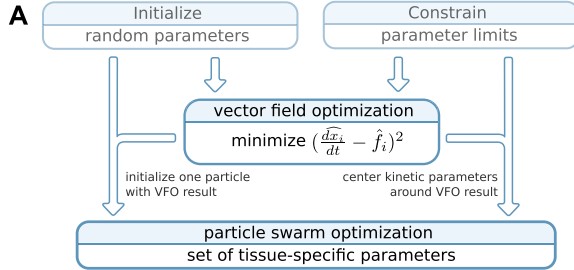

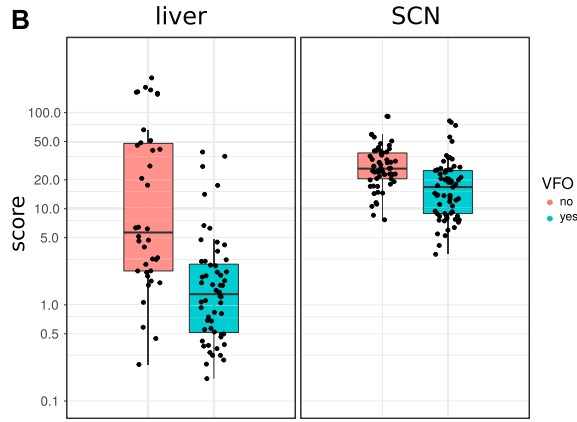

**Figure 4. Improved fitting with VFO.**
**(A)** Flow diagram showing how VFO is integrated into the fitting procedure. The resulting parameter set is used to initialize one particle and to pre-emphasize kinetic parameters. **(B)** Score for fits to circadian transcription data from mouse liver and SCN with and without VFO. Each point represents a fitted model. VFO leads to significantly lower score values (Wilcoxon rank–sum test, *P*-value liver: $4.29 \times 10^{-7}$, *P*-value SCN: $4.819 \times 10^{-6}$).

There was not just a single optima of global optimization, but for all investigated tissues, multiple parameter configurations fitted the data.

To determine essential feedback loops for each model fit, we use our clamping protocol published in 2016 (13). Clamping of genes is carried out by setting the expression level of genes to their mean value (constant) and corresponds to constitutive expression experiments in the wet laboratory (36, 40, 41, 42). It allows comparison of the effect of rhythmic versus basal regulation.

In addition to gene clamping, we also clamp specific regulations via gene products. In silico, this is carried out by setting the corresponding terms in the differential equations of the model constant. Regulations/terms are shown as network links in Fig 1. We can examine the relevance of feedback loops associated with such links by clamping regulations systematically.

To reduce computational effort, we use a targeted clamping strategy, testing specifically which feedback loops are essential. We regard a negative feedback loop as essential for oscillations if clamping of each link that is part of the loop disrupts rhythmicity (only one link at a time is clamped). Details are provided in Supplementary Information 5.

In addition, we test the synergy of loops by clamping combinations of regulations. We are able to distinguish two different modes of synergistic function: (i) two loops work independent of each other and mutually compensate for perturbations, and (ii) two dependent loops share the required feedback for oscillations, such that rhythms only occur when both loops are active.

For example, in the liver, we found many model fits in which the *Bmal1–Rev-erb-α* and repressilator loops function synergistically. Both loops constitute a negative feedback from *Rev-erb-α* onto itself and are timed accordingly, such that they mutually support rhythmicity.

We apply this clamping analysis to every generated model fit (420 in total; Table 1) to identify feedback loops responsible for rhythmicity. The most commonly found loops are marked in Fig 1.

### Loop and parameter composition reflects variation of clock gene expression

For every tissue-specific data set, we have obtained multiple model fits. Along the lines of the previous studies (43, 44), we exploit the ensembles of tissue-specific data sets to extract characteristic model properties for each organ. After assigning essential loops to every model fit, we are able to compare variations in loop composition between different tissues.

Therefore, we define four core loops that were identified by our analysis and discussed in the literature. Fig 5 shows how the composition of these four loops varies between tissues. Interestingly, there are marked differences.

In liver, *Bmal1–Rev-erb-α*, *Per2*, *Cry1* loops, and repressilators all occur with comparable frequencies and, thus, appear to fit the data equally well. In contrast, the SCN repressilators only occur in a few cases. This is also consistent with the early peak time of *Cry1* mentioned earlier because the repressilator mechanism is based on distinct inhibitions at different phases (13). Fits to the adrenal gland and kidney data have similar proportions of essential loops, and in contrast to the liver, they have more *Bmal1–Rev-erb-α* loops and less repressilators. The profiles of additional tissues are shown in Supplementary Information 5.

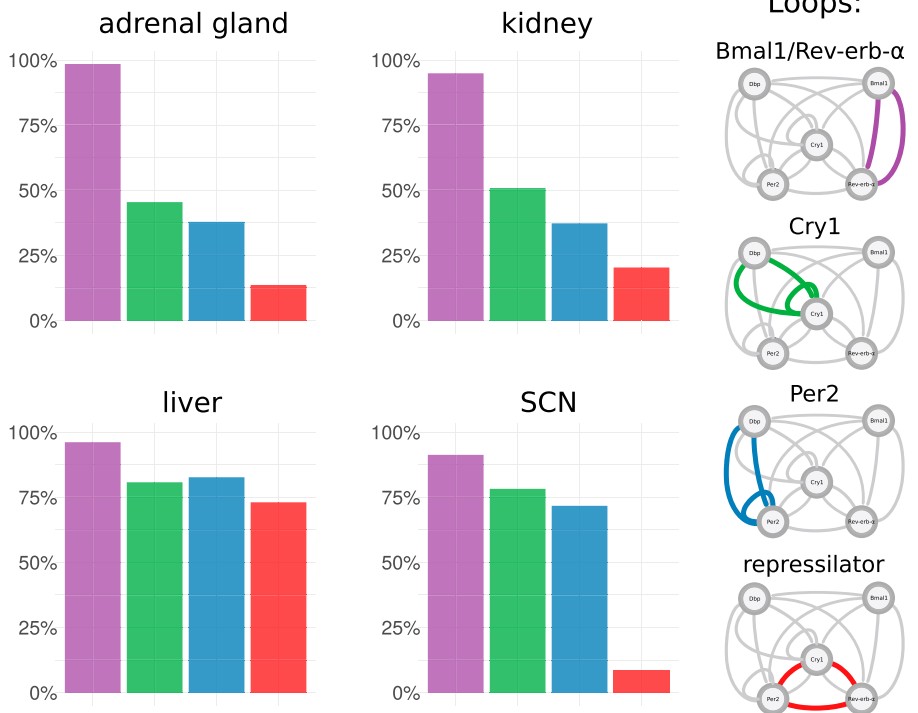

**Figure 5. Proportions of essential feedback loops across tissues.**
For each tissue-specific transcriptome data set multiple model fits were generated. In each model fit, essential feedback loops were then identified using clamping analysis. Frequencies are shown for a set of four core loops that were most prominent in the analysis and are discussed in the literature.

To find out whether tissue differences are reflected in the model parameters, we examine their distributions in the 34-dimensional parameter space. To this end, we perform dimensionality reduction by principal component analysis and visualize tissue differences using linear discriminant analysis (45).

Fig 6A illustrates that the parameters sets fitted to SCN are clearly different from those fitted to other tissues. The differences can be assigned to selected parameters as indicated by the red arrows.

In Fig 6B, we project the model parameters to the first two principal components and color the points according to the essential loops. It turns out that *Per2* loops (blue), *Cry1* loops (green), and *Bmal1–Rev-erb-α* loops (orange) are associated with distinct parameter sets. We observe, for example, an association of *Cry1* loops with high *Cry1* delay.

The differences in loop distributions (Fig 5) and parameter constellations (Fig 6) suggest that differences between expression profiles (Fig 2) imply tissue-specific mechanisms to generate self-sustained oscillations. For example, in brain tissues, small amplitudes and early *Cry1* phases promote self-inhibitions of *Per2* and *Cry1*, whereas a large *Rev-erb-α* amplitude in liver leads to many solutions with *Bmal1–Rev-erb-α* loops and repressilators.

### Synergies of feedback loops

So far, we discussed tissue-specific frequencies of single loops. Interestingly, most parameter sets cannot be assigned to unique loops but to combinations of different essential feedback loops. Now, we use a targeted clamping strategy (Supplementary Information 5) to explore possible synergies of feedback loops.

Our clamping strategy allows us to find loops that are necessary (or essential) for rhythm generation. If we clamp regulations that are

part of these loops, rhythms vanish. Furthermore, by clamping many regulations at the same time, we can also identify sets of loops that are sufficient for oscillation generation. In simulations, rhythms persist if these loops are active while all others are clamped. We term such synergistic sets of loops "rhythm-generating oscillators."

Analyzing 420 parameter sets, we find more than 70% that exhibit synergies of different feedback loops. Fig 7A illustrates that most models constitute combinations of loops. Venn diagrams in Fig 7B show that in liver, *Bmal1–Rev-erb-α* loops together with repressilators form the largest group of oscillators, whereas in the SCN, *Bmal1–Rev-erb-α* loops are typically associated with *Per2* and *Cry1* loops.

Interestingly, the synergy of multiple loops leads typically to low scores. There is a significant difference in the number of loops between parameter sets greater than and lower than the median score (Wilcoxon rank-sum test, P-value < 0.0001) and most model fits involving all four loops lead to excellent scores lower than 2.5.

Moreover, repressilators exhibit quite good scores, in particular, for the liver, kidney, and adrenal gland. For these tissues, fits with repressilator have an average score of 1.24, whereas fits without repressilator have a mean score of 4.41. This is consistent with our finding that better scores involve more loops. The repressilator motif connects inhibitions of *Per2*, *Cry1*, and *Rev-erb-α* and links the loops synergistically.

## Discussion

Circadian rhythms in mammals are generated by a cell-autonomous gene regulatory network (46). About 20 regulators drive core clock genes via E-boxes, D-boxes, and RREs (10). Based on clustered gene expression phases (compare Fig 2), we reduced the system to a network of five genes connected by 7 positive and

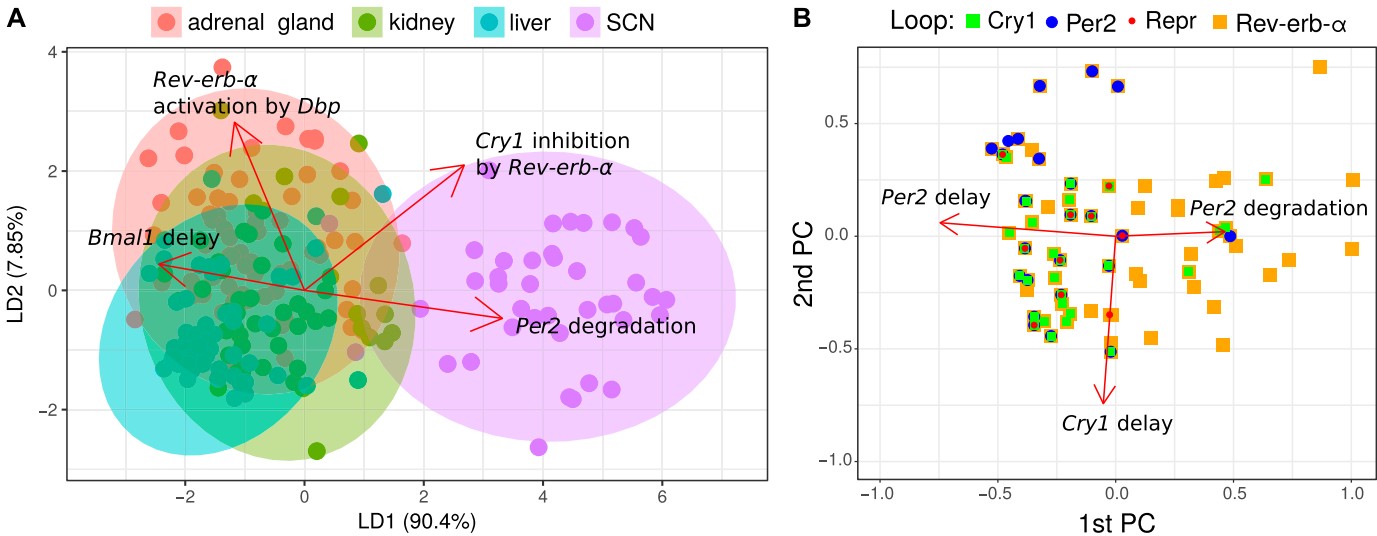

**Figure 6. Tissue-specific models separated in parameter space.**
**(A)** Linear discriminant analysis. Fits (points) are projected to a plane while trying to maximize the variance between tissues. The projected parameter vectors are visualized as arrows in this plane, showing how parameters differ between tissues. Only the four largest arrows are shown for simplicity. **(B)** Loops in parameter space. Shown are the first two principal components and points corresponding to parameter sets for the adrenal gland. Directions of the parameter axes are given as red arrows. Relations between parameter values and loops are visible, for example essential *Cry1* loops (green) occur, when *Cry1* delays are large. Only the three largest arrows are shown, which are markedly larger than the others.

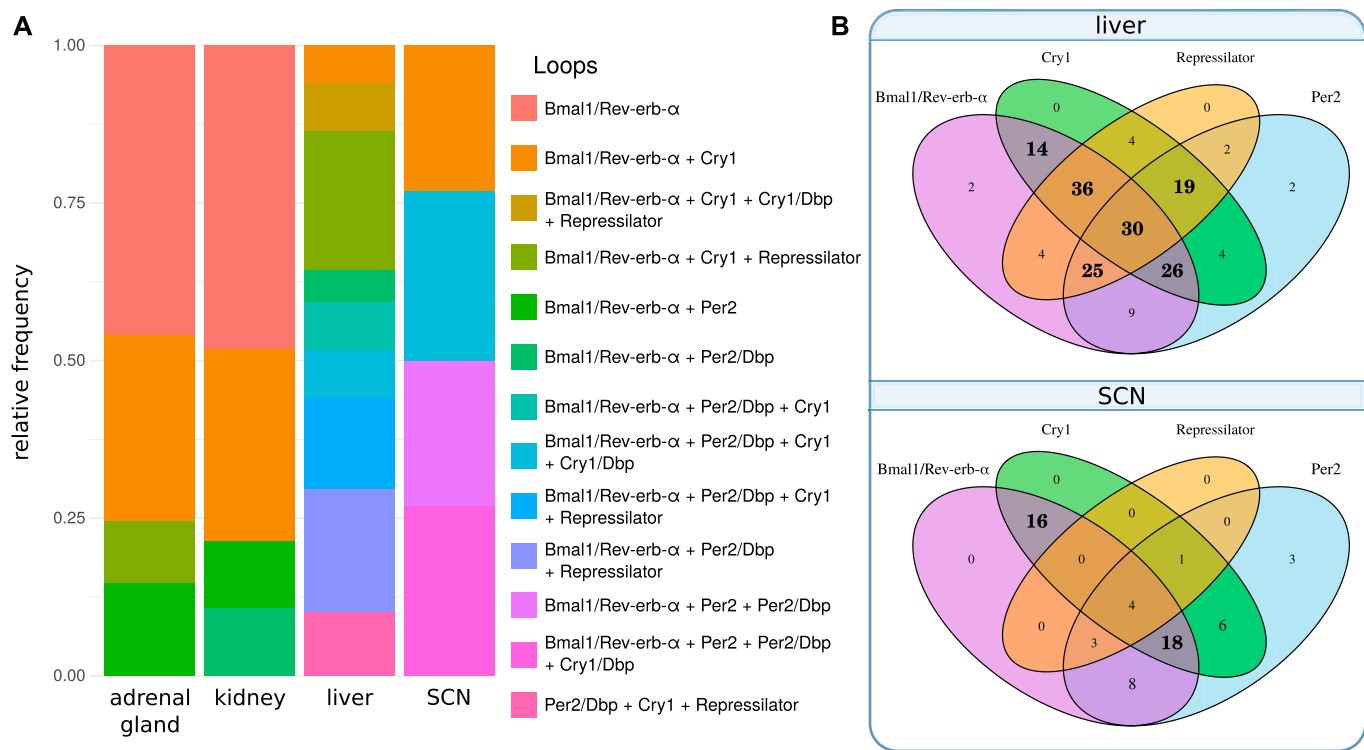

**Figure 7. Proportions of minimal oscillators across tissues.**
**(A)** Frequencies of oscillators in models of different tissues. An oscillator comprises one or more loops (connected with a "+" in the legend). *Per2*, *Cry1* loops, and their extensions via *Dbp* (Fig 1) are counted separately here. **(B)** Venn diagrams of oscillator composition for liver and SCN. Bold numbers highlight the most frequent subsets. In liver, most oscillators comprise many loops including *Bmal1–Rev-erb-α* loop, repressilator, *Per2*, and *Cry1* loops. In the SCN most oscillators are a combination of *Bmal1–Rev-erb-α* with *Cry1* and *Per2*.

10 negative regulations (Fig 1). This reduced model still contains multiple positive and negative feedback loops.

Our aim was to identify the most essential feedback loops and to quantify tissue differences. Our network model was fitted to comprehensive expression profiles of 10 mammalian tissues (3). Furthermore, we use proteomics data (22), ChIP-seq data (11, 23), and decay-rate data (25) to constrain the ranges of unknown parameters. Since quantitative data on protein dynamics are sparse, we simplified the model by using explicit delays between transcription and regulation.

We optimized parameters by a combination of a novel approach termed VFO and PSO (38). After combining these global optimization techniques, our simulations could reproduce the data within experimental uncertainties.

To our surprise, we found for every studied tissue multiple excellent fits with quite different parameter constellations. To extract the responsible feedback loops we performed a systematic clamping analysis. Individual regulatory terms (edges in the network) were systematically clamped to constant values. These clamping methods revealed the essential feedback loops in each of the networks derived from tissue-specific expression profiles.

We found an astonishing diversity of essential feedback loops in models that were able to reproduce the experimental data. Among the essential loop structures we found *Per* and *Cry* self-inhibitions. These loops have been considered as the primary negative feedacks because the double knockouts of *Cry* genes (47)

and the triple knockouts of *Per* genes (48) were arrhythmic. Later, additional feedback loops via nuclear receptors have been found (49). As predicted by modeling (17) and confirmed by *Rev-erb* double knockouts (18), the *Bmal1/Rev-erb* loops constitute another possible rhythm generator. Indeed, in all tissues, we detected parameter constellations that use this negative feedback loop.

Recently, the repressilator, a chain of serial inhibitions, was suggested as a possible mechanism to generate oscillations in the liver and adrenal gland (13). This loop structure is associated with dual modes of E-box inhibitions (30) based late *Cry1* expression (35) and late CRY1 binding to E-boxes (11). Because the expression phase of *Cry1* is tissue-dependent, it is plausible that the detection of repressilators also might differ between different organs.

Indeed, repressilators are less frequently detected as essential in models based on brain data–derived parameter sets as shown in Fig 5. In general, the model parameters in the SCN are clearly different from the parameters in peripheral tissues (Fig 6). Thus, modeling can point to different design principles in specific organs. The large amplitudes and late *Cry1* phases in tissues such as liver suggest that the repressilator is a relevant mechanism in these tissues, whereas small amplitudes and early *Cry1* phase in brain tissues favor *Cry* and *Per* self-inhibitions.

Major differences between tissues have been reported also regarding amplitudes and phases of clock-controlled genes (3, 4, 5, 6). Such differences are presumably induced by tissue-specific

transcription factors (34, 50, 51). Moreover, different organs receive different metabolic and neuroendocrine inputs leading to quite different rhythmic transcriptomes. These systemic tissue differences can also modify the core clock dynamics. In particular, nuclear receptor rhythms differ drastically between organs (52) and can induce tissue specificities of the core clocks (53, 54).

Interestingly, the best scoring models involve several essential feedback loops. This observation indicates that the synergy of different feedback mechanisms improves oscillator quality. Furthermore, co-existing loops imply redundancy, and thus, the core clock is buffered with respect to non-optimal gene expression, hormonal rhythms, seasonal variations, and environmental fluctuations.

Co-occuring feedback regulations might also explain reports where different circadian outputs displayed slightly different periods. For example, in SCN slices, different reporter signals indicated distinct periods (55, 56), and also in crickets, two independent negative feedback loops were reported (57). In some of our high-scoring networks, we indeed find two independent frequencies leading to slight modulations of the circadian waveforms (Supplementary Information 6).

Tissue-specific core clock mechanisms are presumably related to functional differences of SCN and peripheral organs. *Per* gene regulations are particularly important in the SCN because light inputs and coupling via vasoactive intestinal peptide induce *Per* genes via cAMP response element-binding protein (16, 58). Relatively

small core clock amplitudes in the SCN allow efficient entrainment and synchronization (59, 60). Moreover, small amplitudes might facilitate adaptation to long and short photoperiods by varying coupling mechanisms (61, 62). The dominant role of *Per* and *Cry* self-inhibitions is also reflected by the arrhythmic activities of *Per* and *Cry* double knockouts (47, 48).

Peripheral organs such as the liver, kidney, and adrenal gland govern the daily hormonal and metabolic rhythms. Consequently, large amplitudes and pronounced rhythms of nuclear receptors are observed (52, 63). Interestingly, we find that feedback loops involving RREs are more prominent in these tissues (compare Fig 8).

We now address the question of how the choice of our simplified model might affect the results. As shown in Table 1, we use many different parameter sets that can reproduce the data within experimental uncertainties. Such a probabilistic interpretation makes our conclusions more robust regarding the choice of parameters. Furthermore, a fit to different qPCR data sets (6) also detected the repressilator motif as a core element in the liver and adrenal gland. The model structure was chosen to be generic and, in particular, not depending on specific molecular mechanisms. It was created in an unbiased way using general assumptions and experimental evidence on interactions. Interestingly, the co-existence of *Per*/*Cry* and *Bmal1*/*Rev-erb-α* loops has been found also in a different larger model (17). Thus, we assume that the main results of our study do not depend much on the model choices made.

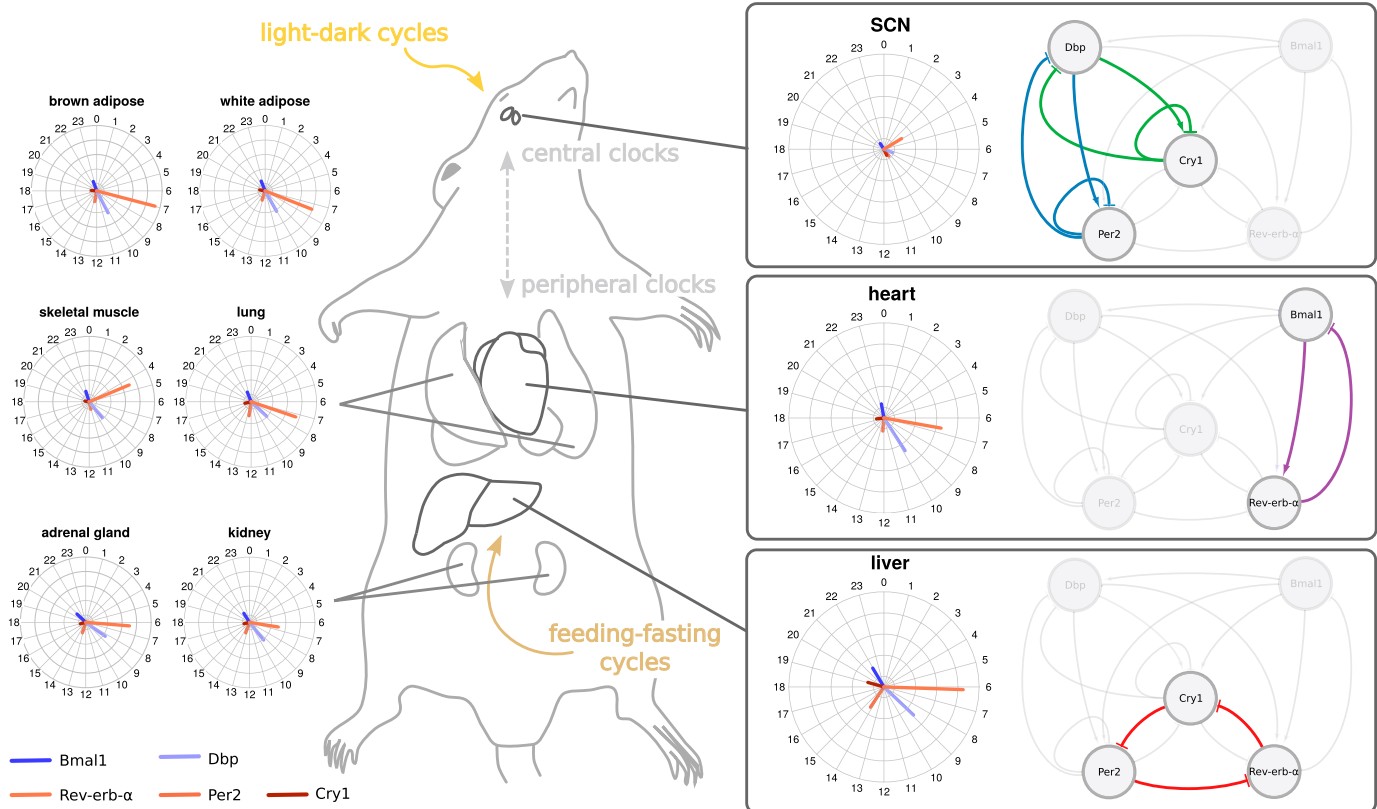

**Figure 8. Differences between gene expression in mouse tissues and associated feedback loops found in model fits.**
Shown are three representative feedback loops for tissues as characteristic examples.

To test our predictions, we suggest tissue-specific modifications of the core clock genes. In particular, constitutive or out-of-phase expression of clock genes can be used. It has been shown already that constitutive expression of *Per* and *Cry* genes impairs rhythms (36, 40, 41). Also, specific regulations can be manipulated experimentally to resemble clamping of regulatory edges. For example, the removal of intronic RREs of the *Cry1* gene leads to vanishing amplitudes in single cells (35). Moreover, available REV-ERB agonists (63) could be applied to study the role of the corresponding loops. Therefore, our model predictions can be tested by specific perturbations resembling our numerical interventions.

In summary, our study suggests that there is not necessarily a single dominant feedback loop in the mammalian core clock. Instead, multiple mechanisms including *Per/Cry* self-inhibitions, *Bmal1/Rev-erb* loops, and repressilators are capable to generate circadian rhythms. The co-existence of feedback loops provides redundancy and can thus enhance robustness and flexibility of the intertwined circadian regulatory system.

## Materials and Methods

Methods can be found in the Supplementary Information.

## Supplementary Information

## Acknowledgements

We are grateful for discussions with Dr. Christoph Schmal, Dr. Bharath Ananthasubramaniam, Dr. Anja Korenčič, and Dr. Matthias König. This work was supportet by grants from Deutsche Forschungsgemeinschaft (HE2168/11-1 and TRR/SFB 186 A16 and A17), Bundesministerium für Bildung und Forschung (01GQ1503), and Graduiertenkolleg CSB 1772/2.

### Author Contributions

JP Pett: conceptualization, data curation, software, formal analysis, validation, investigation, visualization, methodology, and writing—original draft, review, and editing.
M Kondoff: conceptualization, methodology, and writing—review and editing.
G Bordyugov: conceptualization, supervision, and writing—review and editing.
A Kramer: supervision, funding acquisition, and writing—review and editing.
H Herzel: conceptualization, supervision, funding acquisition, writing—original draft, project administration, and writing—review and editing.

### Conflict of Interest Statement

The authors declare that they have no conflict of interest.

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
