## [Reviewer comments · Life Science Alliance]

Co-existing feedback loops generate tissue-specific circadian rhythms

J. Patrick Pett, Matthew Kondoff, Grigory Bordyugov, Achim Kramer and Hanspeter Herzl
DOI: 10.26508/lsa.201800078

Review timeline:	Submission date:	24 April 2018
	1 st Editorial Decision:	17 May 2018
	1 st Revision received:	1 June 2018
	Accepted:	4 June 2018

Report:

(Note: Letters and reports are not edited. The original formatting of letters and referee reports may not be reflected in this compilation.)

1st Editorial Decision

17 May 2018

Thank you for submitting your manuscript entitled "Co-existing feedback loops generate tissue-specific circadian rhythms". Your manuscript was peer-reviewed by three experts and you can find their reports below.

As you will see, all three reviewers appreciate your study and support publication of it in Life Science Alliance. We would thus be happy to publish your paper in Life Science Alliance pending minor revision to incorporate the suggestions made by the reviewers and to meet our formatting guidelines.

REFeree REPORTS

Reviewer #1 (Comments to the Authors (Required)):

The authors use parameter fitting of a simplified model of the mammalian circadian clock to examine the regulatory structure of clocks in the SCN and peripheral tissues. They use global optimization techniques to fit mathematical models to circadian gene expression profiles for different mammalian tissues. Essential feedback loops differ between tissues, pointing to different structures of the clock in different tissues.

The modelling work is well done and will be of interest to readers, and the technique of parameter optimization will also be of interest. My only major point is that it would be useful to have a bit more discussion about how the choice of the 5-gene oscillator model, although well justified from Figure 2, might effect the results, and what experiments could be done to test the differences between tissues suggested by the modelling approach. Otherwise the manuscript is well written, thoughtfully laid out, and justifies its conclusions well.

Reviewer #2 (Comments to the Authors (Required)):

In this article, Pett et al. present core clock network and how multiple feedback loops differ between the tissues with specific design principles within the hierarchy of mammalian tissue. More importantly, how the functional feedback loops increase the robustness and flexibility of the

circadian core clock in mammalian tissue. Pett et al. extended their previous work (Pett et al 2016) where they explored systematically circadian oscillators model with multiple negative and positive feedback loops, identifying a key design principle repressilator as a core element of the mammalian circadian oscillator. This paper is particularly geared to grasp and understand the dynamic nature of complex feedback regulations between the organs. The results are interesting and model predictions are compared with experimental data. To this end, the authors created a core clock model with a 5-gene regulatory network containing most essential loops (cf. Figure 1). Further, authors translated the core clock network into 5-delay differential equations with 34 parameters (cf. S1). The authors used in combing vector field and particle swarm optimization approach to obtain the parameters. Here, the author used published data of 10 different tissue-specific expression profiles such as adrenal gland, kidney, liver, heart, skeletal muscle, lung, brown adipose, white adipose, SCN and cerebellum for model optimization. The significant parameters were chosen based on scoring functions (cf. S3). The model and the data fitting was shown nicely in (cf. Figure 3A, 4B). Later, authors used a clamping method (Pett et al 2016) to determine the essential feedback loops for each model fit as well as to understand the necessity of the loops structure for rhythm generation. Interestingly, authors found that tissue-specific parameters set to reproduce the experimental data best; also the essential loop structures vary between the organs (cf. Figure 5). Among the essential loop structure, authors found that Per and Cry gene shows self-inhibitions also considered as a primary negative feedback loop, this loop predominate in SCN clock model while in Liver models many loops act in synergy and are connected by repressilator motif. Similarly, authors found that in Liver repressilator motif dominant whereas Bmal1/Rev-erb- α loops are found in the heart. As per model prediction and verified by experiment that Bmal1/Rev-erb - loop, and repressilator can generate circadian rhythm in mammalian tissue. All above findings clearly show that different regulation of loop structure in different tissue. As discussed in the manuscript itself, the core clock dynamics (amplitudes and phases) differ from one tissue to another because of their tissue-specific transcription factors expression. This experimental/theoretical effort tackles a general problem of integration of complex feedback regulations between the organs. Such theoretical effort must be pursued to add new predictions and make the model even more useful.

Reviewer #3 (Comments to the Authors (Required)):

This new study by Pett et al. examines tissue-specific circadian oscillations by fitting a simplified circadian clock model to gene expression data derived from various different mouse tissues. Their model is "simplified" in the sense that similarly-regulated genes are often captured by a single exemplar and many cellular processes are captured through explicit delay-differential equations. At the same time, it is still complex enough to contain multiple feedback loops, each of which could sustain oscillations in certain parameter regimes.

For each tissue-specific dataset, they fit their model multiple times using a global optimization algorithm. In each case, they found that a wide range of parameters fit the model acceptably, rather than convergence to a single global minimum. This phenomenon has been observed in similar modeling studies, including those of the circadian clock (e.g., Jolley et al., *Biophysical Journal* 107(6):1462-1473).

My personal favorite step in their analysis was when they "clamped" the expression of various clock components at constant levels in order to determine which feedback loops were really essential. They found that model fits of similar quality produced not only wide variations in parameter values, but also diversity in which feedback loops were really essential to maintaining oscillations. At the same time, the dominance of different feedback loops varied across tissues, with Bmal1/Reverb-A elements dominating in the adrenal gland and kidney, and Cry1 and Per playing a larger role in the SCN.

This paper makes a strong case for probabilistic interpretations of models. Rather than a single "correct" set of parameters -- or even a single correct interpretation -- they offer an ensemble of model interpretations for each tissue. In my opinion, probabilistic interpretations of simple models like theirs are likely to be a good guide for understanding the broad strokes of what drives clock function in different tissues. I strongly recommend publication.

A few minor comments on specific sections of the paper:

Pg. 7: It wasn't immediately clear (without consulting the supplementary information) what it means for an optimization run to have a score of less than 10. Supplementary Section S3 explains (if I'm interpreting it correctly) that a score of 10 is derived from tolerance values, which in turn come from an analysis of the likely experimental error. It might be good to mention in the main text that optimizations were scored on their ability to reproduce the correct period, phases, and fold change, and that the cutoff of 10 reflects a fit that is within the experimental noise. Figure 3A is really helpful in this respect, giving a sense of what an 0.01-fit and a 3.36-quality fit look like. An example with a fit score of 10 would be nice to see as well.

Pg. 8-9: I had to read the paragraph introducing vector field optimization a couple of times before I understood what was going on. I think this is because, after reading the preceding paragraph on PSO, I was picturing time series of the particle swarm in parameter space, not the model-space time series derived from experimental values. Maybe this would be clearer if you rephrased "available time courses for each variable" as something like "experimentally-derived time courses for model variables." Or you could introduce VFO before you introduce PSO and eliminate any chance for confusion.

Pg. 13, Figure 6: I found the red arrows in these figures a little hard to interpret, especially in panel A (where there are a lot of them). Maybe this could be complemented with a table that would include each parameter, its meaning (i.e., which element is being delayed, degraded, etc.) and its interpretation (larger in SCN, etc.). Also, it seems there were 34 different parameters, but I'm seeing a lot fewer arrows (especially in panel B, with only 3). Is this because the other 31 parameters made negligible contributions to the first two principal components? If so, that would be a useful result to mention, as it highlights that only a few parameters have systematic differences in different tissues or oscillation types.

1st Revision – authors' response

1 June 2018

Reviewer #1:

"My only major point is that it would be useful to have a bit more discussion about how the choice of the 5-gene oscillator model, although well justified from Figure 2, might effect the results, and what experiments could be done to test the differences between tissues suggested by the modelling approach."

A1: First, the Reviewer asks us to extend the discussion to address the influence of the model choice on our results. To this end we add a paragraph at the end of the discussion, commenting on the generality of our approach and relating to other work that supports our main results.

A2: Second, the reviewer requests a discussion of experiments that could be done to test our model prediction. We add another paragraph at the end of the discussion in which we suggest tissue-specific clock gene modifications, such as constitutive expression experiments.

Reviewer #2

We are pleased about the reviewer's positive remarks and affirmation. Also we hope to address the reviewer's last remark that "*Such theoretical effort must be pursued to add new predictions and make the model even more useful.*" by our added discussion on the generality of our model predictions and possibilities of experimental verification, as also requested by Reviewer #1. Our methods published in this work are also ready to be applied in future research to additional data to add new predictions.

Reviewer #3

Q1: The reviewer highlights the advantage of "probabilistic interpretations of models" and points to other work where the phenomenon of wide parameter ranges that fit the data similarly well has been observed previously.

A1: We add a sentence at the beginning of section 2.4 to mention these results.

Further the reviewer lists a few minor comments:

Q2:

"It wasn't immediately clear (without consulting the supplementary information) what it means for an optimization run to have a score of less than 10."

A2: We add a sentence in the third paragraph of section 2.2 to explain the interpretation of a score of 10. Also, we add an example of a score 10 fit to supplement S3 as further suggested by the reviewer.

Q3: In the paragraph introducing vector field optimization:

“Maybe this would be clearer if you rephrased “available time courses for each variable” as something like “experimentally-derived time courses for model variables.”

A3: We followed the reviewer’s advice and rephrase the sentence as suggested.

Q4:

“Figure 6: I found the red arrows in these figures a little hard to interpret, especially in panel A (where there are a lot of them). [...] Also, it seems there were 34 different parameters, but I’m seeing a lot fewer arrows (especially in panel B, with only 3). Is this because the other 31 parameters made negligible contributions to the first two principal components?”

A4: We edit Figure 6, reduce the amount of arrows and add more meaningful labels at the arrow heads. Indeed, only a subset out of 34 parameter-arrows is plotted for the sake of readability. We only plot the top 4 and 3 most contributing parameters in Figure 6A and B respectively. There is no clear cutoff for Figure 6A in the data. Therefore we pick a number of 4 that allows easy visual inspection. For Figure 6B there is a gap in the arrow-contributions to the fourth largest arrow. Therefore we pick the top 3 largest arrows for display. However, it is not possible to conclude that contributions of other parameters are completely negligible.

Thank you for submitting your Research Article entitled "Co-existing feedback loops generate tissue-specific circadian rhythms". I appreciate the response to the reviewers you provided, and I am happy to accept your manuscript for publication in Life Science Alliance. Congratulations on this interesting work.